# Assessment of Public’s Awareness Regarding Irritable Bowel Syndrome in Aseer Region, Saudi Arabia

**DOI:** 10.3390/healthcare11081084

**Published:** 2023-04-11

**Authors:** Mohammed A. Bawahab, Muneer Jan Bhat, Fahad Nasser Mohammed Asiri, Khalid Ali Mohammed Alshahrani, Abdulaziz Mohammed Alshehri, Bassam Ahmed Almutairi, Muath Mohammed Alhumaidi, Rayan M. Eskandar

**Affiliations:** 1Surgery Department, Faculty of Medicine, King Khalid University, Abha P.O. Box 641, Saudi Arabia; 2Ibn Sina Medical College, Jeddah P.O. Box 641, Saudi Arabia

**Keywords:** irritable bowel syndrome, knowledge, risk factors, general public, Saudi Arabia

## Abstract

Irritable bowel syndrome (IBS) is a gastrointestinal disorder characterized by altered bowel habits, abdominal pain, or discomfort. It is a highly prevalent disorder that affects patients’ quality of life. A workup is usually required to diagnose IBS, as its differential diagnosis includes some serious conditions such as carcinoma of the colon. The present study aimed to assess the awareness and beliefs of the general population regarding IBS. This study was conducted in the Aseer Region, in the southwestern part of Saudi Arabia. It followed a cross-sectional research design that was conducted during the period from January to March 2021 using a structured self-administered questionnaire to assess the demographic variables in addition to questions to assess participants’ awareness and beliefs related to IBS. Following a convenience sample, the study included 779 participants, with 43.3% being male, mostly in the age group 21–30 years (36.7%), and 68.7% being university graduates. Most participants (70.5%) were aware of IBS, and had the correct knowledge about its etiology, symptoms, risk factors, prognosis, and management. It is recommended to conduct various awareness-raising programs regarding IBS to improve the public’s knowledge and to decrease functional disabilities and their impact on life.

## 1. Introduction

Functional gastrointestinal disorders (FGID), such as functional dyspepsia (FD) and irritable bowel syndrome (IBS), are featured by chronic or recurrent abdominal symptoms of pain. In the case of irritable bowel syndrome (IBS), it is connected with either relief or exacerbation by defecation, or changes in bowel habits [1]. In addition, changes in bowel habits are related to both relief and exacerbation. Mood disorders, intestinal microbiota, chronic infections, altered intestinal permeability, and low-grade mucosal inflammation, including eosinophilia, systemic immune activation, altered bile salt metabolism, abnormalities in the serotonin metabolism, and genetic factors have all been proposed as potential causes of FGID [1].

Irritable bowel syndrome, often known as IBS, is a condition that affects the digestive tract and is characterized by stomach pain and chronic discomfort, as well as changes in the appearance of stools, mucus discharge, and bowel abnormalities [2,3]. IBS is not caused by any medical illness. It is a chronic and debilitating functional disorder of the gastrointestinal tract that affects 9–23% of the population across the world, causing a considerable impact on the quality of life, and resulting in functional disabilities [3]. According to the findings of a study that was carried out in the Al Jouf district of Saudi Arabia, the prevalence of IBS ranges between 8.9% and 9.2%. The incidence of irritable bowel syndrome (IBS) is considerably higher in persons under the age of 40 compared to those beyond the age of 40 [4].

The diagnosis of IBS is a matter of debate. Most community health providers believe that IBS is a diagnosis of exclusion; however, the guidelines emphasize that IBS is not a diagnosis of exclusion, and they encourage clinicians to make a positive diagnosis using the Rome criteria alone [5,6]. One of the most important differential diagnoses of IBS is colon cancer [7]. IBS and colon cancer share similar symptoms. Some patients with colon cancer may be misdiagnosed as having IBS. Nevertheless, with colon cancer, a person may experience unexplained weight loss and blood in their stool from the rectum. These symptoms do not occur in IBS. Despite similar symptoms, IBS does not put a person at a higher risk of developing colon cancer [8]. IBS frequently presents with serious symptoms that may be even more frequent than diabetes or hypertension [9]. However, the treatment of IBS is difficult because of its diversity and complexity. Although there are various guidelines related to its management, the main focus remains on the efficacy of medications using high-priority endpoints, leaving those of lower priority to be largely unreported.

There is poor knowledge about its prevalence, precipitating factors, and associated factors in Saudi Arabia [10]. To the best of our knowledge, there has been no study evaluating the knowledge of the general population regarding IBS. Thus, it is highly essential to have the basic knowledge and awareness of related symptoms and management of IBS. Hence, with this background, this study was conducted aiming to assess the knowledge, and related factors regarding IBS among the general population of the Saudi community in Aseer Region.

## 2. Methods

### 2.1. Study Design

The present study followed a cross-sectional research design. The study population included all Saudi adults (aged >18 years). However, those with a history of IBS, or who worked in the healthcare field were excluded. Following a convenience sample, a total of 779 participants could be included during the period from January to March 2021.

### 2.2. Data Collection Tool

Based on a thorough review of the relevant literature, a self-administered structured questionnaire was constructed by the researchers to assess participants’ personal characteristics (i.e., age, gender distribution, body mass index, marital status, residence, employment profile, and educational level), in addition to well-constructed questions regarding participants’ knowledge and attitude about IBS.

The study questionnaire’s content validity was assessed by an expert panel comprising two physicians and one research consultant. Moreover, the Cronbach α-coefficient was applied to measure its internal consistency, which proved to be appropriate (α = 0.84).

### 2.3. Data Collection

The study was conducted in accordance with the Declaration of Helsinki and was approved by the local ethics committee of the institute (ECM#2020-2402). Informed written consent was obtained from all subjects prior to their enrollment in the study. Patients were clearly informed about the questionnaire and the responses to all questions were recorded and analyzed.

The questionnaire was shared as an online questionnaire via various social media applications such as WhatsApp, Facebook, Twitter, Instagram, and Telegram, so as to achieve maximum responses from the general public and cover all types of education groups and subjects with a varied demographic profile. Moreover, face-to-face data collection was also performed by trained medical students in the form of direct interviews of the general population visiting public places, such as shopping malls and parks.

### 2.4. Outcomes

We evaluated participants’ personal characteristics (i.e., age, gender distribution, body mass index, marital status, residence, employment profile, and educational level) and their knowledge and attitude regarding IBS.

### 2.5. Statistical Analysis

All data were attained and recorded in a pre-designed and validated Excel sheet. Collected data were statistically analyzed using the Statistical Package for Social Sciences (IBM, SPSS, version 20.0). Descriptive statistics (i.e., mean, standard deviation, frequencies, and percentages) were computed. Testing the significance of differences was performed using the chi-square test. The level of significance was set at a *p*-value < 0.05. Testing the significance of differences was performed using the chi-square test. Binary logistic regression was applied to identify the independent variable significantly associated with public awareness of IBS. The level of significance was set at a *p*-value < 0.05.

## 3. Results

A combined total of one thousand people were invited through the use of social media and personal invitations. There were 822 responses that were accurate, which equates to 82.2%. Notwithstanding this, more participants with IBS and healthcare staff, totaling 43 patients, were excluded from the study. The total number of respondents was 779, which equates to a response rate of 77.9%.

Table 1 shows the personal characteristics of participants, with 43.3% being males. The age group of 36.7% was 21–30 years, while 8.6% were aged < 20 years. Most participants (79.2%) were living in the Aseer Region, Saudi Arabia, while 20.8% were temporary visitors to the area. More than half of the participants (58.0%) were married. More than two-thirds of the participants (68.7%) were university graduates, while 20.3% were secondary school qualified. Almost one-third of the participants (32%) were students, while 27.9% were teachers. More than one-third of the participants (39.5%) were overweight, while 24.1% were obese.

Figure 1 shows that most participants (70.5%) were aware of IBS, while 29.5% were not aware of it.

Table 2 shows participants’ responses regarding their awareness and knowledge of IBS. Almost half of the participants (49.4%) became aware of IBS from their friends or relatives, followed by the Internet and social media (36%). Almost one-third of participants (31.5%) stated that the most common symptoms of IBS include bouts of diarrhea, constipation, and abdominal cramps, while 60.8% stated that IBS is due to a colon infection.

More than half of the participants (54.6%) were not aware of the prevalence of IBS, while 28.8% correctly stated that IBS was more common than diabetes or hypertension, whereas 16.7% denied that. More than three-quarters of participants (79.5%) indicated that IBS affects the quality of life, while 92.3% stated that it could cause rapid mood changes. More than half of the participants stated that abdominal pain and excess gas are the most irritating symptoms of IBS that make the patient uncomfortable (62.4% and 56.4%, respectively), followed by constipation (36.6%), and diarrhea (8.1%). Almost half of the participants (48.9%) stated that dietary habits or food allergies are the most common etiological factors affecting patients with IBS, followed by genetic factors (38.1%), anxiety/depression/stress, infection (31.2%), and motility problems (1.5%). More than half of the participants stated that IBS can lead to ulcers and cancer (53.1% and 45.3%, respectively), followed by absorption disorders (30.4%), malnutrition (20.9%), and hemorrhoids (17.6%). A combination of multiple factors was stated by 43.6% of participants to be the most common triggering factor for IBS, followed by stress (31.2%), frequent and excessive intake of meals (16.3%), excessive intake of coffee (13.4%), excessive eating of meals containing saturated fats (5.5%), and eating spicy foods (5.1%). Most participants stated that dietary changes and medications could improve IBS symptoms (80% and 71.6%, respectively), whereas knowledge about the role of the surgery was not particularly known by most participants (70.6%). Most participants (76.6%) stated that counseling a physician could significantly improve the IBS status. The impact of advanced age was not well known by participants (46.2%), whereas only 21.8% confirmed its impact, and 32% denied its impact.

About one-third of participants (38.8%) stated that they advocate consulting a gastroenterologist for a relative who seeks advice regarding IBS, followed by recommending a change in diet (37.6%) or offering herbal medications (16.4%). About two-thirds of participants (67.7%) wished to know more about IBS, followed by which foods are to be avoided (11.7%), what causes IBS (4.6%), and lastly medications to prevent IBS attacks (0.1%).

Table 3 shows that participants’ awareness of IBS was significantly less common among females (*p* = 0.004), those aged more than 40 years (*p* < 0.001), married (*p* < 0.001), those with less than a university educational level (*p* = 0.037), and overweight and obese participants (*p* = 0.005).

Table 4 shows the binary logistic regression model for predictors of participants’ awareness of IBS. The odds ratio, i.e., Exp(B) for the significant independent variables was 6.4 for the participants’ gender, 5.37 for their age groups, 4.49 for their qualification, 3.69 for their marital status, and 3.54 for their body mass index. All the independent variables included had a statistically significant association with participants’ awareness of IBS (*p* < 0.001).

## 4. Discussion

The disorders of brain–gut interactions are a category of functional gastrointestinal disorders that are defined by a complicated relationship between the brain and the gut. They are also known as DGBIs. These conditions are characterized by a number of symptoms, including pain in the abdominal region, bloating, diarrhea, and constipation, all of which can have a major influence on a person’s quality of life [11]. Increasing pieces of evidence point to the possibility that alterations in the microbiota of the gut, the function of the intestinal barrier, and the immune system may all play a role in the pathogenesis of DGBIs. In addition, the impact that stress, worry, and depression has on the gut–brain axis have the potential to make the symptoms of DGBIs worse [12]. Recent developments in our knowledge of DGBIs have resulted in the creation of new diagnostic and therapeutic approaches for the treatment of these illnesses [13]. These include the use of probiotics, prebiotics, and dietary modifications in order to improve the composition and function of gut microbiota. Additionally, psychological therapies such as cognitive behavioral therapy and mindfulness-based interventions are utilized in order to target the psychological components of DGBIs [14]. In addition, there is evidence to suggest that novel pharmacological treatments, such as inhibitors of selective serotonin reuptake and certain neuromodulators, may have therapeutic benefits in the treatment of DGBIs [15]. In general, these new methods show promise for boosting the management of DGBIs and the quality of life of those who are afflicted by them.

Irritable bowel syndrome is a prevalent, commonly occurring, functional chronic gastrointestinal disorder that carries a considerably negative impact on social functioning and quality of life; thus, the condition should not be underestimated [16]. It is observed that during a person’s lifetime, one in every five persons suffers from IBS. Moreover, the number of persons taking medical advice regarding IBS is also increasing, which accounts for around 12% of the total patients visiting primary health care clinics. This group constitutes a major part of patients reporting to clinics of the gastroenterology department [10].

Anxiety and stress are known to make symptoms of irritable bowel syndrome (IBS) worse, which makes it more difficult to control the disease [17]. Anxiety and stress are known to trigger the release of hormones, such as adrenaline and cortisol, both of which have the potential to irritate and inflame the digestive tract [18]. In addition, anxiety and stress can upset the balance of bacteria in the gut, which can make the symptoms of irritable bowel syndrome (IBS) worse. Alterations to one’s diet, which can be brought on by anxiety and stress in certain people, might make the symptoms of irritable bowel syndrome (IBS) worse [19].

The incidence of irritable bowel syndrome (IBS) continues to vary greatly across the globe. The prevalence of irritable bowel syndrome (IBS) is significantly higher in females than it is in males. Individuals between the ages of 20 and 40 years old were affected the most frequently [10]. The prevalence of irritable bowel syndrome (IBS) is estimated to range anywhere from 9–40%, according to the findings of a number of studies that have been carried out all over the world [20,21,22]. Insufficient data are available regarding the prevalence of irritable bowel syndrome (IBS) and the factors that put people at risk in Saudi Arabia’s general population. Studies have also proposed that the incidence of IBS can also be affected by various genetic factors, where family history plays an important role in the development of IBS in around 30% of patients [22].

Most of the studies have focused on the prevalence of IBS among different populations [10,23,24], rather than the awareness and knowledge about IBS among the general public. The present study was performed as the first study to evaluate the public’s knowledge about IBS and its risk factors in the Aseer Region, Saudi Arabia. The questionnaire used in the study was validated and revised to focus on knowledge about the cause, nature, risk factors, and management of this syndrome. Thus, this study did not replicate the questionnaires or data of the previous published.

The etiology of IBS remains unknown, lacking confirmed or known factors. It has been advocated that the pathophysiology of IBS is not yet properly understood. For explaining the pathogenesis of diseases, various linked risk factors have been proposed. Recently, a “biopsychosocial” model has been proposed in an endeavor to harmonize and integrate various factors (such as environmental, genetic, and psychological) that work together synergistically to generate different signs and symptoms [24].

According to the findings of this study, most of the adults in the Aseer Region of Saudi Arabia are familiar with irritable bowel syndrome (IBS), and the vast majority of these individuals have accurate knowledge regarding the disorder’s etiology, symptoms, risk factors, prognosis, and management, particularly among university-educated subjects. On the other hand, the results of the bivariate and multivariate analyses suggested that knowledge of IBS was considerably lower among participants who were obese, younger, married, and had a lower level of education.

However, the study by Hakami et al., which aimed to assess the knowledge, attitude, and practices of medical students in Saudi Arabia toward IBS, reported that most participants had misconceptions about its etiology and nature [25]. Similarly, several other national [26,27] and international studies [28,29,30] reported low awareness of IBS among the study populations, as most of the subjects had improper knowledge about the etiology and risk factors of IBS and its symptoms.

The present study also revealed that the general public has adequate knowledge about the risks and aggravating factors of IBS. Most subjects will seek medicinal treatment and professional counseling for the management of IBS. Similar findings were reported in several studies [27,28,29], whose participants advocated the need for awareness about IBS and various factors related to the disease.

### Strengths and Limitations

The present study was the first to explore the knowledge and related risk factors for IBS among the general public in Aseer Region, Saudi Arabia. However, this study has a few limitations, such as the cross-sectional nature of the study design, and the non-probability sampling used to include participants, which may reflect a selection bias. Moreover, the majority of participants were of young and middle age, and most respondents were highly educated, which may reflect some degree of response bias. In addition, the study did not inquire about the relatives of the participants who may suffer from IBS. Further studies should be conducted on a larger sample size, involving different parameters and variables to study the knowledge, attitude, and awareness among study subjects in relation to IBS.

## 5. Conclusions

Most of the adult Saudi general public has good knowledge about the etiology, symptoms, risk factors, prognosis, and management of IBS. However, further observational studies need to be conducted with an elaborated sample size that includes an equal distribution of subjects of different age groups, genders, and educational levels from different areas of Saudi Arabia to assess the knowledge and awareness regarding IBS and the actual prevalence of the disease among the general population in Saudi Arabia. It is also recommended to conduct various healthcare programs to enhance knowledge and raise the public’s awareness about IBS.

## Figures and Tables

**Figure 1 healthcare-11-01084-f001:**
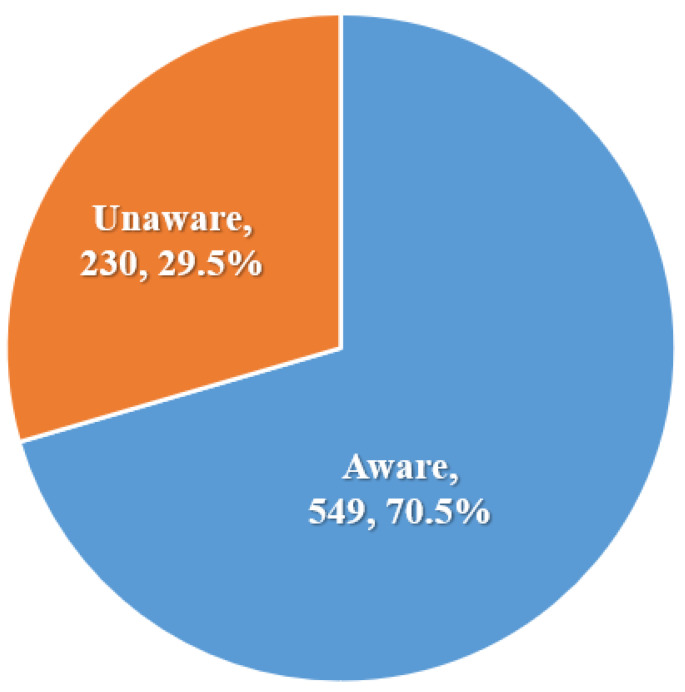
General public’s awareness regarding irritable bowel syndrome.

**Table 1 healthcare-11-01084-t001:** Demographic data (*n* = 779).

Parameters		No. of Subjects	Percentage
Residence	Within Aseer Region	617	79.2
	Outside Aseer Region	162	20.8
Gender	Male	337	43.3
	Female	442	56.7
Age (years)	<20	67	8.6
	21–30	286	36.7
	31–40	180	23.1
	≥40	246	31.6
Marital status	Married	452	58.0
	Single	327	42.0
Education level	Illiterate	2	0.3
	Primary (finished 6 years of school education)	13	1.7
	Intermediate (finished 9 years of school education	27	3.5
	Secondary (finished 12 years of school education)	158	20.3
	University	535	68.7
	Postgraduate	44	5.6
Job	Student	249	32.0
	Teacher	217	27.9
	Administrative	13	1.7
	Military	21	2.7
	Not employed/housewife	168	21.6
	Govt employee	12	1.5
	Retired	35	4.5
	Private sector employee	42	5.4
	Other	22	2.8
Body mass index (kg/m^2^)	Underweight	42	5.4
	Normal weight	241	30.9
	Overweight	308	39.5
	Obese	188	24.1

**Table 2 healthcare-11-01084-t002:** Participants’ responses to knowledge items related to IBS.

Parameters		Options	No. of Subjects	Percentage
Have you ever heard about irritable bowel syndrome?	Yes (*n* = 555, 71.2%)	The Internet and social media	200	36.0
		Competent person/specialized facility	76	13.7
		Relatives or friends	274	49.4
		Others	6	1.1
	No		224	28.8
2. Irritable bowel syndrome includes the following:	Diarrhea/constipation/abdominal cramps		245	31.5
	Colon infection		474	60.8
	Change in appetite		14	1.8
	Ulcers in the bowel		46	5.9
	Inflammation of the bowel		0	0
3. Irritable bowel syndrome is more common than diabetes or hypertension.	True		224	28.8
	False		130	16.7
	I do not know		425	54.6
4. Do you think Irritable bowel syndrome affects the quality of life?	Yes		619	79.5
	No		39	5.0
	I do not know		121	15.5
5. Irritable bowel syndrome can lead to rapid mood changes.	Yes		719	92.3
	No		8	1.0
	I do not know		52	6.7
6. What do you think is the most irritating symptom that makes the patient uncomfortable? (more than one option is allowed)	Abdominal pain		486	62.4
	Excess gases		439	56.4
	Diarrhea		63	8.1
	Constipation		285	36.6
7. Irritable bowel syndrome is a result of the following: (you can choose more than one)	Genetic factors		297	38.1
	Infections		137	17.6
	Dietary factors/food allergy		381	48.9
	Anxiety/depression/stress		243	31.2
	Motility problem		12	1.5
8. Irritable bowel syndrome can lead to the following: (you can choose more than one)	Cancer		353	45.3
	Ulcer		414	53.1
	Malnutrition		163	20.9
	Absorption disorders		237	30.4
	Hemorrhoid		137	17.6
9. Irritable bowel syndrome attacks/episodes are triggered by the following: (you can choose more than one)	A combination of multiple factors		340	43.6
	Eating spicy foods		40	5.1
	Frequent and excessive intake of meals		127	16.3
	Eating a lot of meals containing saturated fat		43	5.5
	By coffee		104	13.4
	By stress at work or in relationships		240	30.8
10. Dietary changes could improve irritable bowel syndrome symptoms.	Yes		623	80.0
	No		18	2.3
	I do not know		138	17.7
11. Prescription medications could improve irritable bowel syndrome symptoms.	Yes		558	71.6
	No		33	4.2
	I do not know		188	24.1
12. Surgery could improve the irritable bowel syndrome status.	Yes		83	10.7
	No		146	18.7
	I do not know		550	70.6
13. Counselling (a gastroenterologist, psychiatrist, or psychologist) could improve the irritable bowel syndrome status.	Yes		630	76.64
	No		26	3.3
	I do not know		600	77.0
14. Advanced age is a major cause of irritable bowel syndrome.	Yes		170	21.8
	No		249	32.0
	May be		360	46.2
15. If one of your relatives has symptoms of irritable bowel syndrome and he/she seeks your advice, what would you do? (more than one option is allowed)	Offer them some herbal medications		128	16.4
	Tell them to change their diet		293	37.6
	Tell them to visit a gastroenterologist		302	38.8
	You will not tell them anything		151	19.4
16. What would you like to know about irritable bowel syndrome?	What irritable bowel syndrome is		527	67.7
	What causes irritable bowel syndrome		36	4.6
	What foods should I avoid		91	11.7
	Medications to prevent attack		1	0.1
	Nothing		124	15.9

**Table 3 healthcare-11-01084-t003:** Participants’ awareness regarding IBS according to their personal characteristics.

		Unaware (*n* = 230)		Aware (*n* = 549)		*p*
Personal Characteristics		No.	%	No.	%	Value
Residence	Outside Aseer Region	53	32.7	109	67.3	
	Within Aseer Region	177	28.7	440	71.3	0.318
Gender	Male	81	24.2	256	75.8	
	Female	149	33.6	293	66.4	0.004
Age (in years)	<20	19	28.0	48	72.0	
	21–30	47	16.4	239	83.6	
	31–40	49	27.3	131	72.7	<0.001
	> 40	115	46.7	131	53.3	
Marital status	Married	166	36.7	286	63.3	
	Single	64	19.6	263	80.4	<0.001
Educational level	Illiterate	1	50.0	1	50.0	
	Primary	5	28.6	8	71.4	
	Intermediate	12	42.9	15	57.1	
	Secondary	59	37.6	99	62.4	0.037
	University	144	26.9	391	73.1	
	Postgraduate	9	21.3	35	78.7	
Body mass index	Underweight	13	31	29	69	
	Normal weight	50	20.7	191	79.3	
	Overweight	102	33.1	206	66.9	0.005
	Obese	65	34	123	66	

**Table 4 healthcare-11-01084-t004:** Binary logistic regression for independent variables affecting awareness of IBS.

		Standard		Exp	*p*	95% CI for Exp(B)	
Variables	B	Error	Wald	(B)	Value	Lower	Upper
Gender	1.86	0.32	34.52	6.40	<0.001	3.45	11.89
Age groups	1.68	0.19	76.92	5.37	<0.001	3.69	7.83
Marital status	1.31	0.37	12.39	3.69	<0.001	1.78	7.63
Educational level	1.50	0.19	62.91	4.49	<0.001	3.10	6.50
Body mass index	1.26	0.18	50.37	3.54	<0.001	2.50	5.02
Constant	−13.72	1.28	115.67	0.00	<0.001		

## Data Availability

The data will be available with the corresponding author to be releases on reasonable request.

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
