# Peer review of "Assessment of Public’s Awareness Regarding Irritable Bowel Syndrome in Aseer Region, Saudi Arabia"

_healthcare, 2023, doi:10.3390/healthcare11081084_

Round 1
Reviewer 1 Report
Overall, the paper is well written and structured. The introduction is relevant with sufficient information about the previous study findings for readers to follow the present study rationale and procedures. The methods are generally appropriate. The results need to addressed some comments. The discussion is
What is the meaning of primary and secondary educational level?
Total number of subjects in the Job section (Table 1. Demographic data) is coming 825, author must crosscheck the figure?
Author must write the option of "you can choose more than one" in the table 2 question no 7 & 16.
Author Response
Dear Chief editor
We would like to thank you and the entire editorial board for your extraordinary efforts in revising and processing our manuscript. We also value the insightful comments of the reviewers, who brought up crucial points that will undoubtedly enhance the quality of the manuscript. The following are our detailed responses to the reviewers' remarks:
Reviewer 1
- What is the meaning of primary and secondary educational level?
- Primary educational level means that the person finished grade 6 in the school education.
- Intermediate educational level means that the person finished grade 9 in the school education.
- Secondary educational level means that the person finished grade 12 in the school education.
- This was added to table 1 as per the reviewer’s suggestion.
- Total number of subjects in the Job section (Table 1. Demographic data) is coming 825, author must crosscheck the figure?
- The number of participants in the study was changed after the exclusion of the IBS patients and the healthcare workers as per the request of other reviewers to become 779 participants.
- Author must write the option of "you can choose more than one" in the table 2 question no 7 & 16.
- The statement “more than one option is allowed” was added to the questions 7, and 16 as per the reviewer’s suggestion in addition to other similar questions. This was corrected in table 2.

Reviewer 2 Report
The manuscript summarizes the results of a survey conducted in general population of Saudi Arabia. The authors attempted to know about the knowledge of participants about irritable bowel syndrome (IBS). I have the follwoingc ommnets
1. How many participants were invited?
2. We shall re-analyze the data after excluding the participants who had IBS because they will introduce positive bias in the results
3. Similarly, health care workers shall also be excluded from analysis
4. At the tiem of data collection the participants, who had their family members with IBS, shall be analysed separately. It may be mentione din discussion as a limitation.
Author Response
Dear Chief editor
We would like to thank you and the entire editorial board for your extraordinary efforts in revising and processing our manuscript. We also value the insightful comments of the reviewers, who brought up crucial points that will undoubtedly enhance the quality of the manuscript. The following are our detailed responses to the reviewers' remarks:
Reviewer 2
- How many participants were invited?
- A total number of 1000 were invited by both social media and by personal invitation. The number of accurate responses were 822 (82.2%). However, further exclusion of the IBS participants and the healthcare workers (43 patients) has been done. So, the final response rate was 779 (77.9%).
- This was added to the results section (First paragraph)
- We shall re-analyze the data after excluding the participants who had IBS because they will introduce positive bias in the results
- The participants who had IBS were excluded from the study. Reanalysis was performed as per the reviewer’s suggestion.
- This was stated and highlighted in the methodology section, results, tables.
- Similarly, health care workers shall also be excluded from analysis
- The healthcare participants were excluded from the study. Reanalysis was performed as per the reviewer’s suggestion.
- This was stated and highlighted in the methodology section, results, tables.
- At the time of data collection, the participants, who had their family members with IBS, shall be analyzed separately. It may be mentioned in discussion as a limitation.
- The study did not inquire about the relatives of the participants.
- This was added to the limitations as per the reviewer’s suggestion.

Reviewer 3 Report
Mohammed A. Bawahab, et al. conducted this study aiming to assess the knowledge, and related factors regarding IBS among general population of the Saudi community in Aseer Region. They demonstrate that almost one third of participants were diagnosed as having IBS, and most of them had correct knowledge about its etiology, symptoms, risk factors, prognosis, and its management. This is an interesting finding and worthy to be explored further. The article can be improved by addressing following concerns.
- Discuss the concept of Disorders of brain-gut interaction.
- Study methods not clear. How was the questionnaire distributed and received? Only by social media and by trained medical students in public places?
- Consider performing multivariate analyses to study which variable is most significant regarding public awareness of IBS.
- Discussion section is very brief. Consider including more data on the impact of social media, mental health issues like anxiety/stress on IBS.
Author Response
Dear Chief editor
We would like to thank you and the entire editorial board for your extraordinary efforts in revising and processing our manuscript. We also value the insightful comments of the reviewers, who brought up crucial points that will undoubtedly enhance the quality of the manuscript. The following are our detailed responses to the reviewers' remarks:
Reviewer 3
- They demonstrate that almost one third of participants were diagnosed as having IBS
- We are sorry for the editing mistake. The participants who suffered from IBS were only 17. And they were removed form the study in order not to confuse our results as per the reviewer’s suggestion.
- Discuss the concept of Disorders of brain-gut interaction.
- The Disorders of brain-gut interaction was added to the discussion section (First paragraph) as suggested by the reviewer
- Study methods not clear. How was the questionnaire distributed and received? Only by social media and by trained medical students in public places?
- More details about the questionnaire distribution were added to the methodology section (last paragraph) as per the reviewer’s suggestion. If the reviewer needs any more details, please clarify and we are fully flexible to add.
- Consider performing multivariate analyses to study which variable is most significant regarding public awareness of IBS.
- Multivariate binary logistic regression analysis was performed, it was added to the results section (table 4).
- Discussion section is very brief. Consider including more data on the impact of social media, mental health issues like anxiety/stress on IBS.
- A paragraph about the impact of anxiety and stress on IBS was added to the discussion section (paragraph 3) as per the reviewer’s suggestion.

Round 2
Reviewer 2 Report
I will thank the authors for accepting my suggestions and making change sin their manuscript
Reviewer 3 Report
Authors addressed my concerns.